# Assessing COVID-19-Related Excess Mortality Using Multiple Approaches—Italy, 2020–2021

**DOI:** 10.3390/ijerph192416998

**Published:** 2022-12-17

**Authors:** Emiliano Ceccarelli, Maria Dorrucci, Giada Minelli, Giovanna Jona Lasinio, Sabrina Prati, Marco Battaglini, Gianni Corsetti, Antonino Bella, Stefano Boros, Daniele Petrone, Flavia Riccardo, Antonello Maruotti, Patrizio Pezzotti

**Affiliations:** 1Statistical Service, Istituto Superiore di Sanità, 00161 Rome, Italy; 2Department of Infectious Diseases, Istituto Superiore di Sanità, 00161 Rome, Italy; 3Department of Statistical Sciences, La Sapienza University, 00185 Rome, Italy; 4Division of Population Register, Demographic and Living Conditions Statistics, Italian National Institute of Statistics, 00184 Rome, Italy; 5Dipartimento GEPLI, Libera Università Maria Ss Assunta, 00193 Rome, Italy

**Keywords:** COVID-19, coronavirus, all-cause mortality, excess deaths, statistical models

## Abstract

Introduction: Excess mortality (EM) is a valid indicator of COVID-19’s impact on public health. Several studies regarding the estimation of EM have been conducted in Italy, and some of them have shown conflicting values. We focused on three estimation models and compared their results with respect to the same target population, which allowed us to highlight their strengths and limitations. Methods: We selected three estimation models: model 1 (Maruotti et al.) is a Negative-Binomial GLMM with seasonal patterns; model 2 (Dorrucci et al.) is a Negative Binomial GLM epidemiological approach; and model 3 (Scortichini et al.) is a quasi-Poisson GLM time-series approach with temperature distributions. We extended the time windows of the original models until December 2021, computing various EM estimates to allow for comparisons. Results: We compared the results with our benchmark, the ISS-ISTAT official estimates. Model 1 was the most consistent, model 2 was almost identical, and model 3 differed from the two. Model 1 was the most stable towards changes in the baseline years, while model 2 had a lower cross-validation RMSE. Discussion: Presently, an unambiguous explanation of EM in Italy is not possible. We provide a range that we consider sound, given the high variability associated with the use of different models. However, all three models accurately represented the spatiotemporal trends of the pandemic waves in Italy.

## 1. Introduction

As of the end of October 2022, over 600 million people worldwide had been diagnosed with SARS-CoV-2 infection, resulting in over 6.5 million reported deaths associated with COVID-19 [1]. WHO data indicate that Italy, the first country to tackle the SARS-CoV-2 epidemic in Europe, was one of the European countries with the highest number of newly reported deaths associated with COVID-19 [1]. While measuring both the direct and indirect impacts of this pandemic on mortality is indisputably crucial for any retrospective and comparative assessment, the most reliable method for achieving this is still debated. Mortality through all causes and excess mortality are the most-used health status-related population indicators for monitoring the impact of any catastrophic event on public health [2]. Excess mortality is defined as the difference between the number of all-cause deaths and the number of expected deaths in the same study period [3]. It includes all effects potentially associated with several factors—often unmeasurable—that could have directly or indirectly affected total mortality. During the first two years of the COVID-19 pandemic, many studies and reports were published worldwide to estimate excess mortality from all causes [4,5,6]. The problem of estimating the expected number of deaths in the absence of the pandemic is highly challenging, and it is still a topic for research. Multiple statistical models and several approaches have been used; the majority have applied time-series-based models [7,8,9,10,11]. Other approaches included Bayesian hierarchical models [12,13]; simple descriptive statistics; and epidemiological approaches [14,15,16,17,18].

We focused on Italy because it was the first western country affected by the pandemic in early 2020 with high excess mortality, especially during the first wave; moreover, Italy was one of the twenty countries accounting for over 80% of the estimated global excess mortality from January 2020 to December 2021 [19]. Several studies have been conducted in Italy in order to estimate excess mortality, especially during the first wave of the COVID-19 pandemic [12,20,21,22]. Some of these analyses provided contrasting numbers during the first epidemic wave (i.e., February to May 2020), with a total excess ranging from 25,700 to 51,000 (ranging from 12% to 24% compared to the reference period, 2015–2019). Such differences were largely due to the different methodologies used, which substantially influenced the estimation of the expected deaths, as pointed out in the UK [23]. Given the increasing importance of excess mortality as an indicator used to quantify the impact of the COVID-19 pandemic, and the high variability in the published estimates, our objective was to compare the results obtained using different models previously applied in Italy. In particular, our estimates were based on the same data set considering data for the first two available years: from January 2020 to December 2021. We highlighted the strengths and limitations of each model and their commonalities with respect to estimating excess mortality during the first two years of the COVID-19 pandemic in Italy.

## 2. Materials and Methods

### 2.1. Data

We obtained data on all-cause mortality in the years 2020 and 2021 and comparable (by week or day) data for earlier years (i.e., 2011–2019) from the National Institute of Statistics (Istituto Nazionale di Statistica, ISTAT). Every year, ISTAT releases daily estimates of the number of deaths by single age, sex, and municipality of residence, drawing information from population and taxpayers’ registries (from the National Register of Resident Population and the information provided by municipalities). The number of deaths registered daily is updated quite regularly (every month) and made publicly available at https://www.istat.it/it/archivio/240401. We downloaded the data used in this study on 18 May 2022. Since July 2020, ISTAT and ISS have periodically published reports in which simple descriptive estimates of excess mortality are estimated as the difference between the number of deaths observed per month and the expected deaths per month, computed as the average number of deaths during the previous five years (2015–2019) [6]. These estimates were adopted as official statistics for the assessment of excess mortality, and, for this reason, we consider them as the benchmark for this study.

### 2.2. Choice of the Models and Description of the Three Models

We chose three models, each representing commonly used approaches for estimating excess mortality: the models created by Maruotti et al. [24], Dorrucci et al. [25], and Scortichini et al. [8]. Some of the coauthors contributed to the implementation of the first two models that have been applied to the same ISTAT data at different spatial resolutions and implemented using the R-language (scripts are publicly available). Further, the three models can be applied at the national and/or regional levels. All three models generate an estimate of excess mortality using different approaches: both Maruotti et al.’s [24] and Scortichini et al.’s [8] models produce a baseline estimate and obtain excess mortality as the difference between the observed number of deaths and the baseline estimates. Conversely, Dorrucci et al.’s [25] model uses the average number of deaths as a baseline and produces an estimate of the excess mortality. The years used for estimating/computing the baseline are also different between the three models: 2011–2019 for Maruotti et al. [24], 2015–2019 for Dorrucci et al. [25], and 2015–2021 for Scortichini et al. [8]. We will refer to the three models as model 1 for Maruotti et al. [24]; model 2 for Dorrucci et al. [25]; and model 3 for Scortichini et al. [8]. The formalization of the three models is as follows.

Model 1: The model is a Negative Binomial GLMM g(E[y|b])=Xβ+Zb, where g is a logarithmic function that defines the relationship between the mean response μ=E[y│b] and the linear combination of the predictors; y is the n × 1 (n = 260 using the years 2015–2019 as a baseline and n = 468 using the years 2011–2019 as a baseline) response vector containing the weekly number of deaths in the baseline years; *b* is a Gaussian random effect vector that corresponds to b~N(0,σ_b); X is an n × 16 fixed-effects design matrix containing vectors defining the 8 Fourier basis functions describing the time dynamic; β is a 16-by-1 fixed-effects vector; and Z is an n × 1 annual random-effects design vector. 

Model 2: Again, the model is a Negative Binomial GLM $g\left(E\left(y\right)\right)=\beta_{0year}+X\beta ,$ where g is a logarithmic function; y is an n × 1 (n = 159) vector containing the weekly (truncated) average number of deaths in the baseline years and the weekly number of deaths in 2020 and 2021; $\beta_{0year}$ is the annual intercept; X is a design matrix containing the b-spline basis function describing the time dynamic; and β is the vector of basis coefficients. 

Model 3: The model is a quasi-Poisson [26] GLM $g\left(E\left(y|X\right)\right)=\beta_{0}+\beta_{1}x_{1}+\beta_{2}x_{2}+\beta_{3}x_{3}+\beta_{4}x_{4}+\beta_{5}x_{5}$, where g is a logarithmic function; y is an n × 1 (n= 2557 using the years 2015–2019 as a baseline and n = 4018 using the years 2011–2019 as a baseline) vector containing the daily number of deaths in the years under study; $x_{1}$ is an n × 1 vector containing the dates; $x_{2}$ is an n × 4 matrix containing a cyclic cubic B-spline with three equally spaced knots for the day of the year; $x_{3}$ is an n × 1 vector containing the day of the week; $x_{4}$ is an n × 6 matrix constrained quadratic B-spline with four equally spaced knots for the days from 1 January 2020 to 31 December 2021 used to define the excess risk in mortality during the COVID-19 outbreak; and $x_{5}$ is an n × 25 matrix defined through a distributed lag non-linear model over 0–21 lag days using a cross-basis parameterization of the mean daily temperature.

Table 1 shows, in brief, the main characteristics of the models.

In this work, we conducted analyses by following the reference articles, maintaining the formalization of the models, the time windows, and the baseline years. Part of the work consisted of extending the time windows until December 2021 and computing the excess mortality estimates using different models on the populations, thereby allowing for comparison. Specifically, we applied the models to the same geographical units, Italian macro-areas (northern Italy, central Italy, and southern Italy/islands), and age groups (0–49; 50–64; 65–79; 80+). Additionally, since the National Health System is organized at the regional level in Italy, we estimated regional models for all age groups together according to the twenty-one regional health systems (more precisely, from North to South, the health-regional systems represented the following regions: Piemonte, Valle d’Aosta, Lombardia, Provincia Autonoma di Bolzano–Alto Adige, Provincia Autonoma di Trento, Veneto, Friuli-Venezia Giulia, Liguria, Emilia-Romagna, Toscana, Umbria, Marche, Lazio, Abruzzo, Molise, Campania, Puglia, Basilicata, Calabria, Sardegna, and Sicilia). We computed the excess mortalities and their confidence intervals for the three models, using different two-time windows to estimate the baseline. We obtained the confidence intervals via bootstrap method for model 1, through parametric point-wise estimation for model 2, and through a Monte-Carlo simulation for model 3. We also calculated the relative excess mortality, called the P-score [27], and discarded the information regarding the confidence intervals. In the following formula, $P_{it}=\frac{ex_{it}}{b_{it}}$, the P-score $P_{it}$ for the i-th region, macro-area, or age group, in the year t-th, is obtained by dividing the excess mortality estimated in the same i-th group and t-th year, $ex_{it}$, for the baseline itself, $b_{it}$.

Given the importance of the baseline in estimating excess mortality, we analyzed possible differences due to changes in the baseline period as follows: 2015–2019 instead of 2011–2019 for model 1, 2011–2019 instead of 2015–2019 for model 2, and 2011–2021 instead of 2015–2021 for model 3.

### 2.3. Cross-Validation Analysis

We validated models 1 and 2 through cross-validation across pre-pandemic years from 2011 to 2019. The cross-validation procedure consisted of two steps: first, we estimated the two models to compute the excess mortality for each year (year of interest) from 2011 to 2019, using the period 2011–2019 (baseline) and removing the year of interest; second, we evaluated these estimates by computing the Root Mean Square Error (RMSE) [28]. This analysis was performed for Italy without stratifying the data by class age, geographical macro-area, or region. The characteristics of model 3 did not allow for cross-validation since the model included a covariate describing the risk of death from COVID-19.

## 3. Results

### 3.1. Excess Mortality Estimates

The number of weekly excess deaths is shown in Figure 1 for the three models: the respective trends were similar for models 1 and 2, highlighting the gap in the observed total mortality during the pandemic years 2020 and 2021, especially in early 2020. Model 3 showed a different temporal pattern of expected deaths with a much less evident distance between the observed total mortality and the baseline, i.e., expected deaths. This was particularly apparent during the peak months.

Table 2 illustrates the results regarding the excess number of deaths estimated by the three models. The excess mortality estimates published in the last ISTAT-ISS report, that is, our benchmark, are also shown for comparison. In detail, Table 2a,b show the raw excess mortality estimates, whilst Table 2c,d show the relative estimates, according to age groups and geographical macro-areas, for the years 2020 (Table 2a,c) and 2021 (Table 2b,d). Models 1 and 3 provided estimates that slightly differed compared to the benchmark, especially in 2021. Overall, only a few counts differentiated model 2’s estimates from the ones reported by the ISTAT-ISS official estimates (+4 death counts). In detail, throughout Italy in 2020, model 1 differs from the benchmark by estimating 558 fewer deaths overall whilst model 3 differs by estimating 15,995 more deaths overall (Table 2 part a). 

In 2021, the same pattern as 2020 is followed: model 2 provided estimates overlapping the ISTAT-ISS official figures, whilst both model 1 and model 3 differed from the benchmark; both model 1 and model 3 showed higher estimates compared to the excess deaths for the age classes 0–49, 50–64, and 65–79. For the age class 80+, model 1 shows a lower estimate compared to the benchmark, while the opposite was shown for model 3. 

Regarding the relative estimates (Table 2c,d), the larger differences with the benchmark were found with model 3, especially for the 0–49 and 65–79 age groups. The discrepancies among the relative estimates across the models were similar to those observed among the raw estimates of excess mortality.

It should be noted that model 1 provides rigorously evaluated confidence bands without using 2020–2021 data in its baseline estimation. For this reason, it returns wide confidence intervals for the excess mortality estimates that mostly include the ISTAT-ISS estimates; the ISTAT-ISS estimate is included in the confidence interval 28 times. The 12 estimates for which model 1’s confidence intervals do not include the ISTAT-ISS estimates correspond to the age groups 0–49 and 65–79.

Appendix A compares the ISTAT-ISS and models 1 and 3’s relative estimates (p-scores). We did not consider model 2 in the graph since it is the most similar to the ISTAT-ISS results. For each geographical macro-area and age class considered in this study, the difference between the models and the ISTAT-ISS model is represented. 

We estimated the values of the same three models at the regional level, producing regional excess mortality estimates. In the Appendix A, we report the results by year and region. All three models’ confidence intervals regarding excess deaths estimated included the ISTAT-ISS official estimates. Differences from the benchmark resulted bigger in smaller regions and in regions with lower mortality. The relative excess mortality estimates (Appendix A) were higher for models 1 and 3 and lower for the ISTAT-ISS report and model 2. The highest relative mortality excesses were found in 2020, mainly in the northern regions (such as Lombardy, Piemonte, Valle D’Aosta, and the Autonomous Provinces of Trento and Bolzano). A visual representation of regional excess mortality has been provided through maps in Appendix A.

**Appendix A.** Excess mortality estimates using different models according to region for the year 2020. The 95% confidence intervals are included for the three models.**Appendix A.** Excess mortality estimates using different models according to region for the year 2021. The 95% confidence intervals are included for the three models.**Appendix A.** Relative excess mortality estimates using different models according to region for the year 2020.**Appendix A.** Relative excess mortality estimates using different models according to region for the year 2021.

### 3.2. Change of Baseline and CROSS-Validation Analysis

Figure 2 shows the total excess mortality estimates for the three models at the national level, using different time windows to compute/estimate the baseline, as explained before. In model 1, the excess mortality estimates were stable for both years regardless of the time window, that is, 2011–2019 or 2015–2019. Whereas the estimates from models 2 and 3 were substantially affected by the choice of the reference period for the baseline estimation. Indeed, with the reference period 2011–2019, the estimates for models 2 and 3 were much higher than those obtained with reference to 2015–2019. The differences found were similar for both years 2020 and 2021. 

The root-mean-square error of model 1 was higher compared with model 2 (804.5 vs. 653.5); again, the latter was largely due to the use of the 2020 and 2021 mortality data in the baseline estimation in model 2.

## 4. Discussion

During the first two years of the COVID-19 pandemic, several studies on excess mortality were published (refer to the introduction). The quantification of COVID-19’s impact on mortality was one of the most debated topics by the scientific community due to its high sensitivity among members of the public. In relation to the official numbers and the majority of the literature, some studies have produced significantly higher estimates [29,30], while others produced the opposite [13,31]. These different quantifications of the pandemic’s impact caused misleading interpretations of the subject; therefore, it is very important to measure the associated uncertainty. This study compared three models with respect to estimating excess mortality in Italy using the same study period, age groups, geographical areas, and mortality data.

To the best of our knowledge, this is the first time that such a comparison has been carried out in order to understand how different methodologies lead to variations in estimates. 

We found different estimates for 2020 and 2021, with significant differences within age groups and geographical areas. We compared each model with the official national estimates [6], which are used by the Italian Government to outline public health policy strategies and interventions. 

In this study, we decided to include the negative values of excess mortality in the results as well, as per the WHO indications: “Negative excess deaths could be observed if deaths that would have happened in the absence of the pandemic were averted due to measures taken to deal with the pandemic”.

### 4.1. Findings

The three models showed a similar temporal trend in excess deaths, reflecting the SARS-CoV-2 epidemic waves since the beginning of the epidemic in Italy, with the highest peak in excess mortality observed during the first months (March and April 2020), followed by a substantial reduction during the summer of 2020 and the start the second epidemic wave in late 2020 [25]. Since March 2021, there has been a decline in excess deaths coinciding with higher vaccination coverage, thereby underlining the effectiveness of the vaccination campaign [32,33]. 

The total excess deaths estimated by year for model 1 and model 2 were quite similar. Both estimates were close to the ISTAT-ISS benchmark, with model 2’s results being almost identical. Model 1’s approach was found to be the most consistent and stable with respect to variations in the baseline years, the population under analysis, and the complexity of the covariates. Model 2 yields results very close to the benchmark, as the model’s structure substantially reflects the simple ISTAT-ISS approach. Further, the adjusted average model quantifies the degree of uncertainty, which is not available in the ISTAT-ISS estimates. 

We found an about 3% difference in model 3’s estimates for both years, making it the one that differs the most from the other two models and the ISTAT-ISS benchmark. Model 3 applied a two-stage design and statistical model that allowed for a flexible estimation of the excess mortality risk and a definition of the baseline risk that considers temporal trends and variations in temperature distribution. These factors seemed to influence the results, especially in 2021, regarding the benchmark and the other two models. Notably, the SARS-CoV-2 epidemic pattern is not typically seasonal in Italy, with the recurring presence of summer epidemic peaks in 2021 [34] and 2022 [35]. 

When estimating the baseline mortality, the influence of historical excess mortality on the parameter estimates, mainly due to heat waves and seasonal influenza epidemics, may play a role in model 1, a point currently under investigation. In this regard, a weighted model could be further considered; accordingly, downweighing historical excess mortality for standardized conditional residuals could be a simple solution [36]. 

Differentiation among the considered approaches emerged when stratifying them by macro-area and age groups. The point estimates became more pronounced for the extreme age groups while the prediction intervals overlapped. Model 1 and model 3’s prediction intervals are wider than that of model 2. 

When comparing the regional estimates in absolute terms, discrepancies were again observed between the three models, especially in regions with an older population and higher expected deaths, for example, the Liguria region in 2021 (Demo-Geodemo—Mappe, Popolazione, Statistiche Demografiche dell’ISTAT).

We found that model 2 was susceptible to the choice of the reference period for the baseline estimation; it produced considerably higher estimates when using a longer reference period. In fact, the non-inclusion of a covariate considering the annual trend in this model likely reflected a non-significant annual trend in the yearly count of deaths in recent years, which is in line with the report by the ISTAT-ISS [6]. This assumption was most likely no longer valid when considering a larger period as a baseline. In contrast, model 1 captures cyclical mortality patterns within years, introducing the Fourier series in the linear predictor. The number of Fourier terms is determined via a model selection procedure applied to the historical mortality data. The variation in the cyclic pattern from year to year is modelled by including random effects, resulting in a mixed-effects model. Notably, it is known that random effects can often represent serial correlation among the measurements. Thus, including random effects may also be sufficient. This assumption was most likely no longer valid when considering a larger period as a baseline. Importantly, it is known that random effects are often able to represent the serial correlation among measurements; thus, including random effects may also be sufficient for capturing the serial correlation between mortality with respect to consecutive weeks. 

Excess mortality has been one of the most used indicators to describe the impact of the COVID-19 pandemic; the early official estimates were based on simple assumptions in Italy [6] and the rest of Europe [37]. Later, many mathematical models were published with the same approaches applied to significantly different contexts and with substantially different estimates than the official ones [29], often disorienting scientific communities [30,38]. Among several publications on excess mortality [12,22,31,39], the estimations for Italy ranged widely, with a great amount of uncertainty, further fueling a national scientific, political, and public debate on the effectiveness of the pandemic response. For this reason, the estimation and comparison of national excess mortality is an extremely sensitive topic, as shown by a recent article in which World Health Organization researchers explain the errors in the high-profile mortality estimates of Germany and Sweden [27]. 

More specifically, our cumulated models’ estimates for Italy in the 2020–2021 period were in the range of 164,000–193,000 excess deaths, largely different from the estimates published in The Lancet by the Global Burden of Disease group in Seattle [29], which estimated an excess of 259,000 excess deaths for Italy. As already mentioned in the literature, the excess deaths estimated by Wang et al. were on average two times higher than those recorded in several high-income countries, including Italy [40]. Including specific covariates in the prediction models may distort this phenomenon [29].

Another study of relevance was conducted by the World Health Organization (WHO) in March 2022 [41]. The WHO’s model differs according to each nation’s availability of all-cause mortality data. For Italy, a Negative Binomial Generalized Additive Model (GAM) was built, with two-time parameters modelled through spline functions. The production of these estimates is a result of a global collaboration between the WHO’s “Technical Advisory Group for COVID-19 Mortality Assessment” and the specific countries. The produced annual estimates for Italy in this case included the ISTAT-ISS baseline within their confidence interval (the global excess deaths associated with COVID-19 (modelled estimates) (who.int)). 

### 4.2. Research Limitations

We limited our choice to only three models to compare excess-mortality estimates, despite the great number of models/analyses applied in the literature [4,5,6,7,8,9,10,11,12,13,14,15,16,17,18]; however, our selection was based on reproducibility, and all three models had already been applied to the Italian official database, which is recorded every month by the official statistics bureau (ISTAT), while two models were based on a time-series approach, which is the most-used method.

No matter which type of modelling is adopted to estimate the excess mortality attributable to this pandemic, the modeling approach will bring about complexity [42]. Many co-factors played a role in determining the observed excess mortality in Italy, namely, a comparatively older population, different regional patterns, and the measures taken by the government during the various stages of the epidemic. However, not all these co-factors are measurable. None of the three models included, for instance, the degree to which the aging Italian population influenced the estimate of the overall excess mortality, and we cannot exclude that this affected the estimates, especially in 2021. An accentuated and prolonged aging phase has characterized Italy for many years. The ISTAT estimates that in 2050, the percentage of people over 65 will rise by more than ten points [6]. Therefore, this factor must be introduced into future models for estimating excess mortality. In future studies on excess mortality, it will be important to discuss the way in which to treat the COVID-19 years with respect to baseline estimations. The pandemic may have modified the structure of the population across the world, and this factor should be considered in excess mortality estimations for future years. A comparison between the excess mortality estimations cited in this article is reported in Appendix A.

## 5. Conclusions

The data and code used in this study are available on Github, and the full set of results is shown in the appendix, making our analysis entirely reproducible. As we mentioned in the methods, we selected those models that allowed us to replicate the results.

An overview of excess mortality in Italy in 2020 and 2021 at different spatial resolutions and focusing on different age classes was presented in this work. Different methods for estimating excess mortality were studied, thus deepening the methodological aspects, even though we have not considered all possible approaches/models for estimating excess deaths. We cannot give an unambiguous answer as to the excess mortality in Italy; nonetheless, our in-depth study of the three models and of other approaches regarding COVID-19-related excess mortality provides a range that we consider trustworthy. It is important to stress that estimates vary greatly depending on the model used in the baseline estimation and its assumptions, which should be made carefully and, above all, verified. 

All mathematical models make several assumptions that should be evaluated, including covariates that are not always significant for the event of interest, i.e., COVID-19-related mortality. This study did not assess the effects of covariates such as temperature; therefore, we cannot assess the way in which this contributed to the difference in the excess mortality estimates observed. While this aspect could have been interesting, it was beyond the objective of this study. What has emerged is that age played an important role, as excess mortality increased corresponding to the age groups in Italy. The latter is expected, given the old age of the Italian population. At the same time, the geographic macro-area showed different behaviors given the different territorial incidences of COVID-19. However, all three models accurately represented the spatiotemporal trends that characterized the pandemic waves in Italy. This work is addressed to all researchers in the field of public health, and is to be used as a reference for future studies on excess mortality

## Figures and Tables

**Figure 1 ijerph-19-16998-f001:**
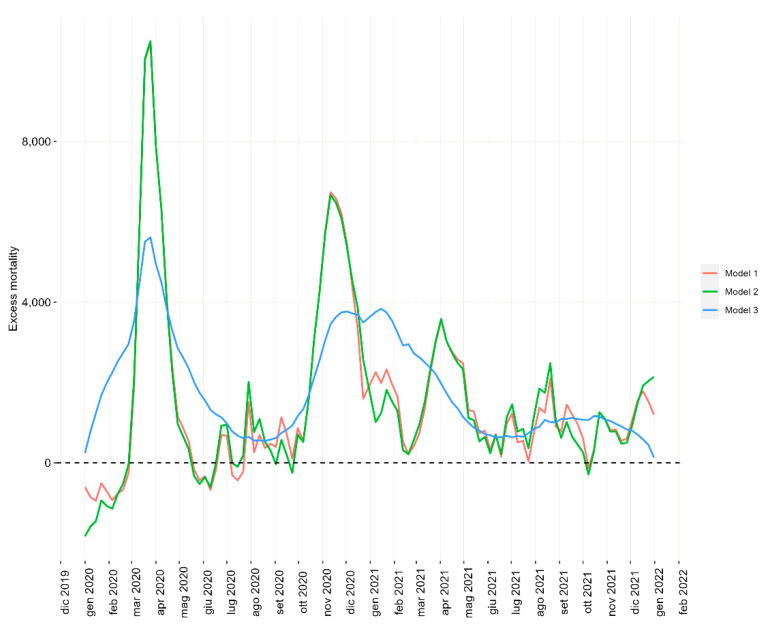
Weekly number of excess deaths estimated by the three models during 2020–2021 in Italy.

**Figure 2 ijerph-19-16998-f002:**
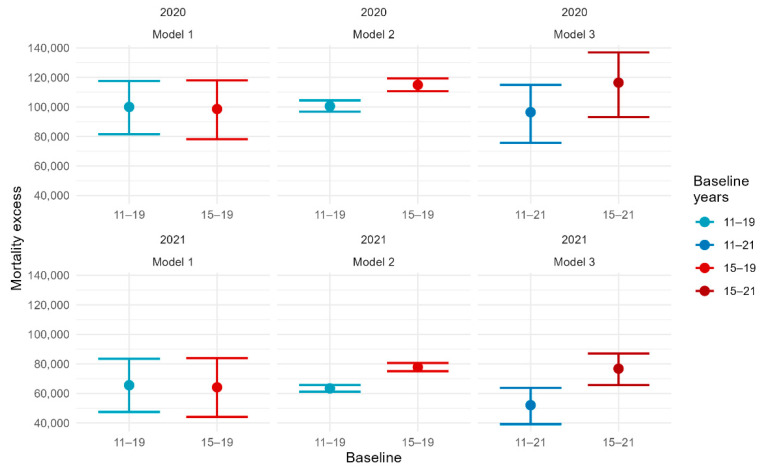
Comparison of the three models estimates using 2011–2019 and 2015–2019 as baselines for the year 2020.

**Table 1 ijerph-19-16998-t001:** Main characteristics of the three models chosen to compare excess mortality estimates during COVID-19 pandemic years in Italy from 2020–2021.

	Model 1	Model 2	Model 3
**Models compared (reference)**	(Generalized mixed effects models; Maruotti A, et al.).	(Generalized additive models; Dorrucci M, et al.).	(Time-series with temperatures distributions; Scortichini M, et al.).
**Statistical model/approach**	Negative binomial mixed model with seasonal patterns.	Negative binomial model/epidemiological approach.	Quasi-Poisson time-series regression model/time-series.
**Type of time modelling and time unit**	Time (weeks) modelled by Fourier series; number of terms chosen by goodness of fit criteria (AIC; BIC).	Time (weeks) modelled by quadratic splines, with one knot per month.	A linear term corresponding to time to model long-term trends, a cyclic cubic B-spline with three equally spaced knots for the day of the year used to model seasonality, and indicators for day of the week to account for weekly variations in mortality.
**Estimate of the number of expected deaths**	Mortality baseline estimated over (2011–2019). The weekly predictions of mortality data for years 2020 and 2021 are based on the 2019 year-specific conditional best linear unbiased predictions of the generalized linear mixed model.	Mean number of deaths during pre-pandemic years (2015–19).	Smooth functions that define a baseline risk accounting for temporal trends and variation in temperature distribution.
**Estimate of excess deaths**	Difference from the estimated baseline along with 95% prediction intervals. If zero is included in the intervals, no difference from the expected number of deaths is hypothesized.	Difference in the number of deaths in 2020/21 adjusted by seasonality with the number of expected deaths.	The excess risk In mortality during the COVID-19 outbreak defined through a constrained quadratic B-spline with four equally spaced knots.
**Strengths of the model**	Simple interpretation and very good in-sample fitting performance are obtained for all Italian regions.	Simple interpretation.	Presence of covariates containing the information regarding mean temperatures.
**Limitation of the model**	Socio-demographic and hospital-related information may improve the accuracy of the estimates and may contribute to explaining heterogeneity across regions. No harvesting effect is considered.	No secular trend estimate.	Low number of cases prevents the full application of the two-stage modelling process in age groups <50 years old.

**Table 2 ijerph-19-16998-t002:** **part a.** Excess mortality estimates for different models according to age and geographical macro-area for the year 2020. The 95% confidence intervals are included for the three models. **part b.** Excess mortality estimates for different models according to the age and geographical macro-area for the year 2021. The 95% confidence intervals are included for the three models. **part c.** Relative excess mortality estimates for different models according to age and geographical macro-area for the year 2020. **part d.** Relative excess mortality estimates for different models according to age and geographical macro-area for the year 2021.

Part a				
	ISTAT-ISS REPORT	Model 1	Model 2	Model 3
**North Italy, 0–49 years**	−552	−149 (−679; 348)	−551 (−564, −539)	195 (−152, 497)
**North Italy; 50–64 years**	3164	3263 (2610; 3882)	3157 (3072, 3252)	4030 (3161, 4789)
**North Italy; 65–79 years**	15,004	17447 (15,,519; 19,275)	14920 (14,231, 15,653)	21051 (18355, 23,322)
**North Italy; 80+ years**	56,681	51,684 (42,501; 60,472)	56,722 (54,362, 59,198)	59,189 (51,678, 65,606)
**Entire North Italy**	74,296	72,695 (63,722; 81,224)	74,241 (71,273, 77,345)	84,507 (73,298, 94,050)
**Central Italy; 0–49 years**	−431	−64 (−371; 216)	−432 (−451, −411)	78 (−102, 232)
**Central Italy; 50–64 years**	392	369 (−16; 728)	389 (381, 402)	796 (375, 1168)
**Central Italy; 65–79 years**	1375	2526 (1513; 3478)	1381 (1339, 1411)	3564 (2408, 4556)
**Central Italy; 80+ years**	8566	7743 (3789; 11471)	8574 (8272, 8888)	9304 (6148, 12,059)
**Entire Central Italy**	9903	10,769 (6480; 14786)	9907 (9588, 10,245)	13,727 (9017, 17,852)
**South Italy and islands; 0–49 years**	−671	−16 (−468; 411)	−673 (−684, −655)	201 (−112, 476)
**South Italy and islands; 50–64 years**	1807	2027 (1480; 2544)	1802 (1764, 1855)	1933 (1162, 2599)
**South Italy and islands; 65–79 years**	3731	4898 (3322; 6383)	3739 (3624, 3866)	6436 (4391, 8215)
**South Italy and islands; 80+ years**	11,461	9260 (3239; 15,001)	11,477 (10,968, 12,005)	11,471 (6698, 15,701)
**Entire South Italy and islands**	16,328	16,588 (10,068; 22,781)	16349 (15,734, 16,995)	20,065 (12,503, 26,896)
**Entire Italy; 0–49 years**	−1654	−183 (−1296; 877)	−1650 (−1679, −1625)	334 (−392, 969)
**All of Italy; 50–64 years**	5363	5767 (4594; 6887)	5362 (5227, 5502)	6600 (4708, 8242)
**All of Italy; 65–79 years**	20,110	25,033 (21,102; 28815)	20,076 (19,300, 20,882)	30,469 (24,724, 35,487)
**All of Italy; 80+ years**	76,708	68,554 (50,239; 86,130)	76,760 (73,682, 79,970)	78,973 (63,764, 92,199)
**All of Italy**	100,526	99,968 (81,474; 117,585)	100,530 (96,805, 104,417)	116,431 (93,146, 136,888)
**part b**				
	**ISTAT-ISS REPORT**	**Model 1**	**Model 2**	**Model 3**
**North Italy; 0–49 years**	−694	−253 (−778; 250)	−699 (−718, −665)	321 (120, 502)
**North Italy; 50–64 years**	1680	1870 (1232; 2501)	1677 (1637, 1731)	2687 (2230, 3102)
**North Italy; 65–79 years**	3866	6683 (4808; 8546)	3862 (3662, 4085)	10,963 (9482, 12,278)
**North Italy; 80+ years**	19,798	15,829 (6723; 24,651)	19800 (18,991, 20649)	20,982 (17,180, 24,455)
**Entire North Italy**	24,649	24,523 (15,682; 33,273)	24,654 (23,656, 25,702)	35,056 (29,546, 40,214)
**Central Italy; 0–49 years**	−315	61 (−241; 344)	−313 (−332, −300)	253 (128, 368)
**Central Italy; 50–64 years**	895	913 (538; 1277)	895 (876, 915)	1482 (1247, 1701)
**Central Italy; 65–79 years**	2030	3296 (2311; 4265)	2030 (1969, 2091)	4878 (4290, 5415)
**Central Italy; 80+ years**	8766	8310 (4430; 12,107)	8768 (8461, 9089)	9072 (7520, 10,530)
**Entire Central Italy**	11,377	12,742 (8586; 16,865)	11,374 (11,001, 11,774)	15,759 (13,471, 17,854)
**South Italy and Islands; 0–49 years**	−339	339 (−111; 770)	−338 (−348, −330)	594 (412, 762)
**South Italy and Islands; 50–64 years**	3227	3543 (3011; 4070)	3227 (3148, 3304)	3253 (2851, 3617)
**South Italy and Islands; 65–79 years**	7443	8842 (7316; 10,349)	7445 (7197, 7694)	10444 (9458, 11,373)
**South Italy and Islands; 80+ years**	17,059	15,335 (9389; 21,136)	17,068 (16,341, 17,820)	13,886 (11,543, 16,057)
**Entire South Italy and Islands**	27,390	28,435 (22,064; 34,742)	27,392 (26,379, 28,453)	28,383 (24,698, 31,975)
**All of Italy; 0–49 years**	−1348	190 (−917; 1249)	−1345 (−1372, −1312)	1069 (688, 1415)
**All of Italy; 50–64 years**	5802	6432 (5283; 7573)	5804 (5663, 5939)	7282 (6368, 8140)
**All of Italy; 65–79 years**	13,339	18,984 (15103; 22,819)	13,340 (12,827, 13880)	25,627 (22,907, 28,172)
**All of Italy; 80+ years**	45,623	39,338 (21216; 57001)	45619 (43,914, 47403)	42,530 (35,147, 49,408)
**All of Italy**	63,415	65,575 (47,496; 83,478)	63421 (61,186, 65750)	76,781 (65,696, 87,079)
**part c**				
	**ISTAT-ISS REPORT**	**Model 1**	**Model 2**	**Model 3**
**North Italy; 0–49 years**	−6.67	−1.91	−6.66	2.59
**North Italy; 50–64 years**	13.97	14.58	13.94	18.5
**North Italy; 65–79 years**	19.9	24.12	19.79	30.36
**North Italy; 80+ years**	28.98	25.99	29	30.66
**Entire North Italy**	24.61	24.15	24.59	28.97
**Central Italy; 0–49 years**	−11.73	−1.95	−11.75	2.46
**Central Italy; 50–64 years**	3.91	3.7	3.88	8.27
**Central Italy; 65–79 years**	4.28	8.22	4.3	11.91
**Central Italy; 80+ years**	9.98	9	9.99	10.93
**Entire Central Italy**	7.52	8.29	7.53	10.74
**South Italy and Islands; 0–49 years**	−8.95	−0.24	−8.98	3.04
**South Italy and Islands; 50–64 years**	9.33	10.68	9.31	10.05
**South Italy and Islands; 65–79 years**	6.53	8.82	6.55	11.83
**South Italy and Islands; 80+ years**	8.95	7.16	8.96	8.95
**Entire South Italy and Islands**	7.7	7.88	7.71	9.63
**All of Italy; 0–49 years**	−8.51	−1.02	−8.49	1.91
**All of Italy; 50–64 years**	10.31	11.25	10.3	12.99
**All of Italy; 65–79 years**	12.22	15.8	12.2	19.75
**All of Italy; 80+ years**	18.73	16.54	18.74	19.39
**All of Italy**	15.57	15.59	15.57	18.49
**part d**				
	**ISTAT-ISS REPORT**	**Model 1**	**Model 2**	**Model 3**
**North Italy; 0–49 years**	−8.39	−3.24	−8.45	4.42
**North Italy; 50–64 years**	7.42	8.35	7.4	12.42
**North Italy; 65–79 years**	5.13	9.24	5.12	16.05
**North Italy; 80+ years**	10.12	7.96	10.12	10.79
**Entire North Italy**	8.17	8.15	8.17	12.03
**Central Italy; 0–49 years**	−8.58	1.86	−8.51	8.14
**Central Italy; 50–64 years**	8.93	9.14	8.93	15.7
**Central Italy; 65–79 years**	6.32	10.72	6.32	16.67
**Central Italy; 80+ years**	10.21	9.66	10.21	10.61
**Entire Central Italy**	8.64	9.81	8.64	12.38
**South Italy and Islands; 0–49 years**	−4.52	4.99	−4.51	9.05
**South Italy and Islands; 50–64 years**	16.67	18.66	16.67	16.83
**South Italy and Islands; 65–79 years**	13.03	15.91	13.04	19.3
**South Italy and Islands; 80+ years**	13.31	11.85	13.32	10.58
**Entire South Italy and Islands**	12.91	13.52	12.92	13.45
**All of Italy; 0–49 years**	−6.93	1.06	−6.92	6.28
**All of Italy; 50–64 years**	11.15	12.55	11.15	14.4
**All of Italy; 65–79 years**	8.1	11.98	8.1	16.83
**All of Italy; 80+ years**	11.14	9.49	11.14	10.31
**All of Italy**	9.82	10.22	9.82	12.14

## Data Availability

Mortality data are publicly available at https://www.istat.it/it/archivio/240401 (accessed on 3 November 2022), and temperature data are available at https://cds.climate.copernicus.eu/cdsapp#!/dataset/reanalysis-era5-land?tab=form (accessed on 14 July 2022).

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
