# Peer review of "Assessing COVID-19-Related Excess Mortality Using Multiple Approaches—Italy, 2020–2021"

_ijerph, 2022, doi:10.3390/ijerph192416998_

Round 1

Reviewer 1 Report

The calculation of excess deaths is very important in assessing mortality and measuring the impact of a pandemic.  As the authors of this paper correctly assert, there are many methods for calculating expected deaths and the result is often variation in the estimates and thus, variation in the estimates of excess deaths.  This paper presents a well-polished evaluation of 3 methods using Italian mortality data.  The analysis is well done and the paper well written.  I have only one suggestion related to negative excess deaths.  Conceptually, when expected deaths exceed observed deaths, there are no excess deaths.  So, one could argue that excess deaths in such circumstances should have a value of zero.  This is important particularly when one is summing excess deaths over demographic categories (e.g., age) and/or geographic areas.  The presence of negative values in one group or area essentially cancels out excess deaths in other groups or areas.  I'm not sure that this is appropriate.  Nevertheless, most of the extant methods ignore this issue.  The authors should at least consider adding a brief note in the discussion about this.  

Reviewer 2 Report

I appreciate the authors’ efforts in the work concerning assessing COVID-19 related excess mortality using multiple approaches, Italy, 2020-2021:

-        The information is relatively easy to navigate, and the structure of the paper allows readers to analyze the concepts approached.

-        The Introduction provides a good background of the domain.

-        The authors bring interesting arguments to the investigated field.

However, to enhance the quality of the study, it would be wise to pay attention to several issues:

(1)   The introduction should clearly state the research question(s) or the research objective(s) that the authors present for this paper.

(2)   The authors should clearly state why did they choose the time frame and Italy as a region of study.

(3)   There is no Literature review section that could bring support to the information provided in the paper. Authors should summarize the opinions of main researchers regarding the topic approached. This could also enhance the reference list, which is currently relatively short.

(4)   Is the mathematical model the contribution of the authors? If not, references must be provided.

(5)   I recommend two different sections to be included in the paper: “Findings”, where authors can present and detail the findings and importance of the study results; and “Research limitations” aside from the Conclusion part.

(6)   The “Conclusions” part must be developed more in order to summarize the main findings of the research, how the research objectives are met through this study and to whom are the results addressed.

Good luck with the revision!

Reviewer 3 Report

Valuable article of high scientific level, especially important for public health professionals.

But some corrections need to be made.

Table 1. Separation of strengths and limitations is not clear. It is better to write them in separate columns. The model’s name could be in a row instead of a column

Maybe the authors could add graphical visualization of excess mortality in Italy using maps? That would help to see differences visually and more clearly.

Table 2.a, Table 2.b., Supplementary Table 1.a, Supplementary Table 1.b, Supplementary Table 2. It is not clear what the numbers in brackets are. The name of statistics (probably 95% confidence interval) must be written.

List of references must be made in order. Links to all online available sources should be added. Incomplete title is written in the reference No.32. The journal, pages and link are missing in the reference No.33.

Round 2

Reviewer 2 Report

The authors took into consideration all the information in the review. The research can be published after these modifications.